# Onion bulb extract can both reverse and prevent colitis in mice via inhibition of pro-inflammatory signaling molecules and neutrophil activity

**Maitham A. Khajah**[1]*, **Ahmed Z. EL-Hashim**[1], **Khaled Y. Orabi**[2], **Sanaa Hawai**[1], **Hanan G. Sary**[2]

1 Faculty of Pharmacy, Department of Pharmacology and Therapeutics, Kuwait University, Safat, Kuwait,
2 Faculty of Pharmacy, Department of Pharmaceutical Chemistry, Kuwait University, Safat, Kuwait

* maitham@hsc.edu.kw, maitham.khajah@ku.edu.kw

**Data Availability Statement:** All relevant data are within the paper and its Supporting Information files.

## Abstract

### Background

Onion is one of the most commonly used plants in the traditional medicine for the treatment of various diseases. We recently demonstrated the anti-inflammatory properties of onion bulb extract (OBE) in reducing colitis severity in mice when administered at the same time of colitis induction. However, whether onion can reverse established colitis or even prevent its development has not been investigated.

### Hypothesis

To test 1. whether OBE can reduce colitis severity when given either before (preventative approach) or after (treatment approach) colitis induction and if so, 2. what are the mechanisms by which onion can achieve these effects.

### Methods

Colitis was induced by dextran sulfate sodium (DSS) administration using treatment and preventative approaches. The severity of the inflammation was determined by the gross and histological assessments. The colonic level/activity of pro-inflammatory molecules and immune cell markers was assessed by immunofluorescence and western blotting analysis. *In vitro* neutrophil superoxide release and survival was assessed by chemilumenecense and Annexin-V/7AAD assays respectively.

### Results

OBE treatment significantly reduced colitis severity in both approaches, the colonic expression/activity profile of pro-inflammatory molecules, inhibited WKYMVm-induced superoxide release, and increased spontaneous apoptosis of neutrophils *in vitro*.

**Funding:** This work was supported by Kuwait Foundation for the Advancement of Sciences (KFAS) (grant number: P11613PT01). Parts of this work were supported by grant SRUL02/13 to the Research Unit for Genomics, Proteomics and Cellomics Studies (OMICS), Kuwait University. We would like to thank Dr. Ananthalakshmi KV for the technical help, the histology unit in the faculty of Allied Health, and the animal unit in Health Science Center, Kuwait University.

**Competing interests:** The authors have declared that no competing interests exist.

**Abbreviations:** DSS, dextran sulfate sodium; IL, interleukin; IBD, inflammatory bowel disease; MAPK, mitogen activated protein kinase; OBE, onion bulb extract; PI3K, phospho-inositol-3 kinase; TNF, tumor necrosis factor; sod, superoxide dismutase.

## Conclusions

OBE can be used as an effective option in the prevention and/or the treatment of established colitis.

## Introduction

Inflammatory bowel disease (IBD) is a bacterially-triggered chronic inflammatory condition of the gastrointestinal tract in a genetically susceptible host [1–4]. Various immune cells (such as lymphocytes, and granulocytes) and inflammatory signaling molecules (such as mitogen activated protein kinase; MAPK family and phospho-inositol-3 kinase; PI3K) play important roles in the disease pathogenesis [5–11]. The currently available pharmacological agents for disease management include corticosteroids (prednisolone, budesonide, etc.), immunosuppressants (cyclosporine, tacrolimus, methotrexate, 6-mercaptopurine), and biological-based therapies (infliximab, adalimumab, vedolozumab). However, most of these agents have limitations in terms of severe adverse effect profile, high cost, and the development of dependence or resistance with chronic use [12–16]. Therefore, the discovery of alternative therapeutic molecules with good efficacy and lower side effect profile would represent a huge advancement in the treatment of IBD.

The use of natural products has gained attraction recently in many disease conditions due to their efficacy and relatively acceptable side effect profile. They showed beneficial effects in various conditions like non-alcoholic fatty liver [17], coronary artery diseases [18], hypertension [19], as well as in gallbladder cancer [20]. *Allium cepa* L. (Family Amaryllidaceae) is considered one of the most commonly used medical plant species in traditional medicine in the management of various conditions including that of inflammatory nature such as asthma [21–24] and recently in IBD [25]. The anti-inflammatory properties of onion are not fully understood but is thought to be mediated, in part, through the reduction of reactive oxygen species, nitric oxide, and pro-inflammatory molecules (tumor necrosis factor; TNFα, interleukin (IL)-1β and -23), as well as increasing the level of the anti-oxidant glutathione [25–28].

We have recently shown, using a murine model of dextran sulfate sodium (DSS)-induced colitis, that prophylactic treatment with onion bulb extract (OBE) ameliorates the gross and histological features of colitis. This was, at least in part, mediated through reducing the colonic expression/activity profile of pro-inflammatory molecules (COX-2, MAPK family, and AKT) and modulating the expression of various pro- and anti-apoptotic proteins [25]. However, in this model, the OBE was administered almost at the same time as the colitis inducing agent, and whilst the effects of OBE were clear, the treatment regimen does not mirror the clinical situation exactly. Clinically, all of the available pharmacological agents are given to patients with established colitis. Indeed, based on various IBD therapeutic guidelines, corticosteroids, and in some cases anti-TNF therapies, are used during the disease flare while other agents (depending on colitis severity and location) such as 5-aminosalycalic acid or azathioprine are used as maintenance therapy. Therefore, the real test for the effectiveness of a natural product (such as onion) is to evaluate its therapeutic efficacy in reducing or reversing an established and active state of colitis.

It is well established that the genetic component is a predisposing factor for colitis and thus constitute high risk factor for its development. Therefore, efforts should be focused on developing preventative strategy approaches, for patients at risk, in order to prevent/delay the disease onset and/or its progression. Therefore, it might be possible that the consumption of high

amounts of onion, within a diet, may decrease the risk of developing colitis or at least lessen its severity. This may necessitate the use of only milder therapeutics agents with acceptable side effect profile (such as 5-aminosalysalic acid) rather than agents with higher side effect profile such as corticosteroids, immunosuppressants, or biological agents. Indeed, there is good evidence that dietary recommendations can be an effective preventative approach, for patients at risk in certain conditions such as; the dietary approach to stop hypertension (DASH) for several cardiovascular conditions such as hypertension and coronary artery diseases. Of interest, it should be noted that, in IBD, there are no current specific dietary recommendations for individuals with risk factors. The only recommendation is to limit dietary intake during the active state of colitis since most of these patients encounter lactose intolerance during the disease flare. Therefore, it is of importance to evaluate if some dietary measurements can be used in the management of IBD as a preventative approach and/or to reduce the disease progression.

In the current study we used the murine DSS colitis model aiming to answer two important questions; 1) if OBE administration can reduce colitis severity when given after establishing a state of active colitis (treatment approach), and/or 2) if OBE would be a useful agent to be given preventatively before colitis induction (preventative approach), which will be helpful as a dietary approach for patients with risk factors for IBD development.

## Materials and methods

### Acquisition of the plant material

Fresh red onion bulbs were obtained through direct purchasing from the local market. The plant was identified as *Allium cepa*, and a voucher specimen, number KOE-010, was deposited at the herbarium of Kuwait University (KTUH), College of Science, Kuwait.

### Extraction of onion bulbs

The extraction procedure was performed as previously described [25]. In brief, about 20 kg of fresh red onion were peeled, coarsely cut and extracted three times, each using 10 L of dichloromethane. The organic layer was separated from the aqueous layer, and the collected organic layers were evaporated in *vacuo* till dryness. The dried extract was dissolved in about 100 mL of dichloromethane and dried over anhydrous sodium sulfate. The produced organic layer was evaporated to dryness in *vacuo* to afford brownish syrupy residue. This extraction process was repeated as needed. The treatment stock solution was prepared using the residue and PBS as a vehicle.

### Animals

BALB/c mice (6–10 weeks old, mean weight 20 g.) were supplied by the Animal Resource Center of the Health Sciences Center at Kuwait University. All animals were kept under standard conditions including controlled temperature (25˚C), a 12-h light-dark cycle and had free access to food and drinking water *ad libitum*. All experimentations were approved by the Animal Care Committee at Kuwait University Health Sciences Center (protocol approval number: P11613PT01) and conformed to their rules and regulations as described previously [29]. During this study, the general health and wellbeing of the mice was continuously monitored and overseen by a veterinary physician in the animal house. All mice were monitored daily for their eating/drinking habits, activity, or other severe signs of hunched or lateral recumbency or starry fur or lethargy. There were no signs of illness or mortality during the treatment time points that required sacrificing the animals.

## Dextran sulfate sodium (DSS) colitis model

Colitis was induced in mice by mixing DSS polymers in the drinking water (3.5% w/v) given *ad libitum* [29, 30]. Control (untreated; UT) mice received tap water only. Mice were divided into the following groups: a) control (UT), b) preventative approach: mice (n = 7-13/group) received daily intra-peritoneal (i.p) injections of OBE or vehicle for 7 days followed by DSS administration (without OBE/vehicle treatment) for 5 days, and c) treatment approach: mice (n = 10-20/group) received DSS for 4 days followed by daily i.p injections of OBE or vehicle for 3 days (without DSS administration). Average daily consumption of water was determined in all groups and was not statistically different. Daily weight changes were determined as loss of baseline body weight; this provides an indirect indication of colitis severity. Mice were sacrificed at the end of the experiment by cervical dislocation and the severity of colitis was determined by macroscopic (gross) and microscopic (histologic) assessments. Colon length and maximal bowel thickness (in millimeters) was also determined [29]. It should be noted that some animals were analyzed for the gross assessment of colitis severity and the colon was subsequently used for other assays such as histological assessment and immunofluorescence analysis.

## Macroscopic (gross) assessment of colitis severity

Using sterile forceps and scissors, the entire colon of each mouse was removed by a ventral midline incision and opened longitudinally. Its length and maximal bowel thickness was measured (in mm) with calipers. The data are presented as the percentage (%) of mice in each group showing various features of colitis (diarrhea, blood in stool, anorectal bleeding, erythema, and edema) [29].

## Microscopic (histological) assessment of colitis severity

Formalin fixed colons were processed for histological assessment, and were blindly scored by 2 observers using a standard semi-quantitative histology scoring system as previously described [29]. The colon was cleaned from stools and blood with a few drops of sterile 0.9% saline and 'swiss-rolled' from the descending to the ascending part. Samples were fixed in 10% neutral buffered formalin and placed in tissue processing and embedding cassettes in PBS for a few minutes and then overnight in a LEICA ASP 3005 tissue processing machine. Tissues were rinsed in two changes of formalin, and then dehydrated in several changes of graded alcohol (70%, 90% and 100%). Three changes of xylene were used for tissue condensation and clearing. Using an SLEE-MPS embedding machine, processed tissues were embedded in paraffin wax (24 h at room temperature) and stored at 4˚C before trimming and sectioning using an LEICA RM 2235 microtome. Sections (6 μm thick) were floated in a water bath and then placed on uncoated slides at 37˚C overnight. After de-paraffinisation in three changes of xylene (5 min each) and rehydration by serial immersion for 2–3 min in each of absolute, 90% and 70% alcohol, sections were washed briefly with distilled water and stained in Meyer's alum haematoxylin solution for 7 min followed by thorough rinsing with running tap water. Before counterstaining sections in eosin solution for 2 min, slides were dehydrated in graded alcohol. A clearing step was also performed by rinsing in three changes of xylene (2 min each) followed by mounting with DPX.

The stained sections were blindly scored by two observers using a standard semi-quantitative histology scoring system which grade the following features: extent of destruction of normal mucosal architecture (0, normal; 1, 2, and 3, mild, moderate, and extensive damage respectively), presence and degree of cellular infiltration (0, normal; 1, 2, and 3, mild, moderate, and transmural infiltration respectively), extent of muscle thickening (0, normal; 1, 2, and

3, mild, moderate, and extensive thickening respectively), presence or absence of crypt abscesses (0, absent; 1, present) and the presence or absence of goblet cell depletion (0, absent; 1, present). The scores for each feature will be summed with a maximum possible score of 11. The extent of ulceration was determined on each section along the muscularis mucosa and expressed as percentage ulcerated mucosa [29].

## Immunofluorescence

Colon sections (5μm) were deparaffinized, rehydrated through a series of washes in graded ethanol and water, followed by an antigen retrieval step (by boiling the sections in 10 mM sodium citrate buffer, pH 6.0 for 20 min). Sections were then incubated in blocking solution (5% bovine serum albumin (BSA) + 0.3% Triton X-100 in PBS) for 1 h, followed by incubation overnight at 4˚C with primary antibodies [p-ERK1/2, p-AKT, p-p38 MAPK, and COX-2) (1:100 dilution)], Cell Signaling, USA, diluted in 1% blocking solution). On the following day, sections were washed and incubated with and the secondary antibody conjugated to Alexa Fluor 555 [Goat anti rabbit SFX kit; Life Technologies, USA (1:400 dilution)] for 2 h at room temperature in the dark. After washes in PBS, sections were stained with 4',6 diamidino-2-phenylindole and mounted. Images were captured on a ZIESS LSM 700 confocal microscope and fluorescence intensity estimated in defined fields using Image J software package [25].

## Western blotting

Colon tissue samples (descending part) taken from naïve (untreated, UT) mice or mice treated with daily DSS plus vehicle or OBE (100 and 200 mg/kg, using the preventative and treatment approach) were cut and homogenized (with Teflon glass homogenizer) in 1 ml of buffer composed of 1.2 g of 50 mM HEPES, 0.3 g of 50 mM NaCl, 1 ml of 0.5 M EDTA, 1 ml of 1% Triton X-100 and 98 ml of deionized water. A protease inhibitor cocktail (10 μg/ml aprotinin, 10 μg/ml leupeptin and 100 μM PMSF) was added separately. Homogenates were centrifuged at 1,800 rpm for 10 min at 4˚C, and the supernatant collected. Protein concentration was determined by the Bradford assay (Bio-Rad, Hercules, CA). Samples containing 40 μg protein were dissolved in an equal volume of 2 x Lammeli sample buffer and β-mercaptoethanol, heated at 90˚C for 10 min and loaded onto a 12.5% SDS-polyacrylamide gel and electrophoresed at 125 V for 1 h. Proteins were transferred (at 100 V for 1 h) onto a PVDF membrane (Millipore, Ireland) and then blocked with 4% BSA for 90 min before overnight incubation at 4˚C with various primary antibodies (1/1000 dilution); p-AKT, p-ERK1/2, T-ERK1/2, p-p38 MAPK, and actin (loading control), and Ly-6G (for neutrophils), F4/80 (for macrophages), CD3ε (for T-cells), and CD19 (for B-cells) (1:200 dilution) (all from Cell Signaling Technology, Boston, MA, USA). Membranes were washed 3 times for 1 h with 1x TBS-T buffer and incubated with appropriate horseradish peroxidase (HRP)-labeled secondary antibodies [Anti-rabbit or Anti-mouse IgG, HRP linked antibody (Cell Signaling Technology, Boston, MA, USA; 1/1000 dilution)]. Bands were visualized using Super Signal ECL substrate (Thermo Scientific, Rockford, USA) and Kodak X-ray film.

## Isolation of bone-marrow derived neutrophils

Neutrophils were isolated from the bone marrow of naïve mice using Percoll gradient as previously described [31–33]. Briefly, mice were euthanized and the femurs and tibias dissected from the animals and the ends of bones removed. Marrow cells were flushed from the bones with ice-cold PBS and collected by centrifugation at 1300 rpm for 6 min at 4˚C. After re-suspension in 3 ml of 52% Percoll (GE Healthcare), the marrow cells were layered on a 3-step Percoll gradient (72%, 64%, and 52% plus cells), and centrifuged at 2600 rpm for 30 min at 4˚C.

Purified neutrophils were removed from the layer between the 64 and 72% Percoll and washed once with ice-cold PBS and then suspended in RPMI containing 20% fetal bovine serum (FBS) at $10^7$ cells/ml. Cytospin of the cell suspension after Percoll gradient purification was performed to identify the cell population as > 95% neutrophils.

## Superoxide release assay

Superoxide anion levels were measured using an assay kit (from Sigma Aldrich) detecting chemiluminescence emanating from oxidation of luminol substrate by released superoxide anions. Freshly isolated neutrophils from naïve mice ($10^6$ cells/100μl) suspended in assay medium were added to reaction mixtures containing assay buffer, luminol, enhancer and superoxide dismutase in triplicate wells of a 96 well plate. Additional wells were prepared with neutrophils alone (vehicle treated) and neutrophils stimulated with WKYMVm in the presence of superoxide dismutase (SOD). Assay buffer and assay medium alone served as a blank for the entire experiment. The reaction was initiated by addition of 10 μM WKYMvm as a superoxide anion inducer. Luminescence intensity was measured immediately and thereafter at 1 min intervals for 30 min using a Thermo Electron Corporation AppliSkan 2.3 Luminometer set at 37˚C in high sensitivity mode.

## Apoptosis assay

Freshly isolated bone-marrow derived neutrophils from naïve mice were re-suspended ($10^6$ cells/ml) in RPMI containing 20% FBS and various doses of OBE (10 ng/ml– 100 μg/ml) or vehicle, plated in 35 mm culture dishes and incubated for 16 h in a 37˚C/ 5% $CO_2$ incubator. Cells were subsequently washed twice (by re-suspension and low speed centrifugation) with ice-cold PBS and once with 1x Annexin-V binding buffer [10 mM HEPES/NaOH (pH 7.4), 140 mM NaCl, 2.5 mM $CaCl_2$]. Pelleted cells were re-suspended in Annexin-V binding buffer and stained for FACS analysis using the PE Annexin V apoptosis detection kit I from BD Pharmingen as per the manufacturer's protocol.

## Statistical analyses

Data were analyzed using GraphPad Instat software (Calfornia, USA). Differences between groups were assessed using a one-way ANOVA followed by Bonferroni post-hoc test, with $p < 0.05$ being regarded as significant.

# Results

The extraction process of OBE was repeated four times to afford 7.5 g (0.038% yield) of brownish syrupy residue.

## Effect of OBE on colitis severity at gross and histological level using the treatment approach

We wanted to determine if OBE treatment modulates the severity of an established colitis in mice. We choose 100–200 mg/kg dose of OBE based on our previously published report using the prophylactic approach [25] which demonstrated anti-inflammatory properties using the same colitis model. Using the treatment protocol (described in Fig 1A), DSS/vehicle treated mice resulted in a significant drop in the body weight (18% from baseline level). In contrast treatment with OBE (100–200 mg/kg) partially prevented the drop in the body weight (Fig 1B). The DSS-induced decrease in colon length (Fig 1C) and the increase in colon thickness (Fig 1D) were also prevented by the OBE treatment (100–200 mg/kg).

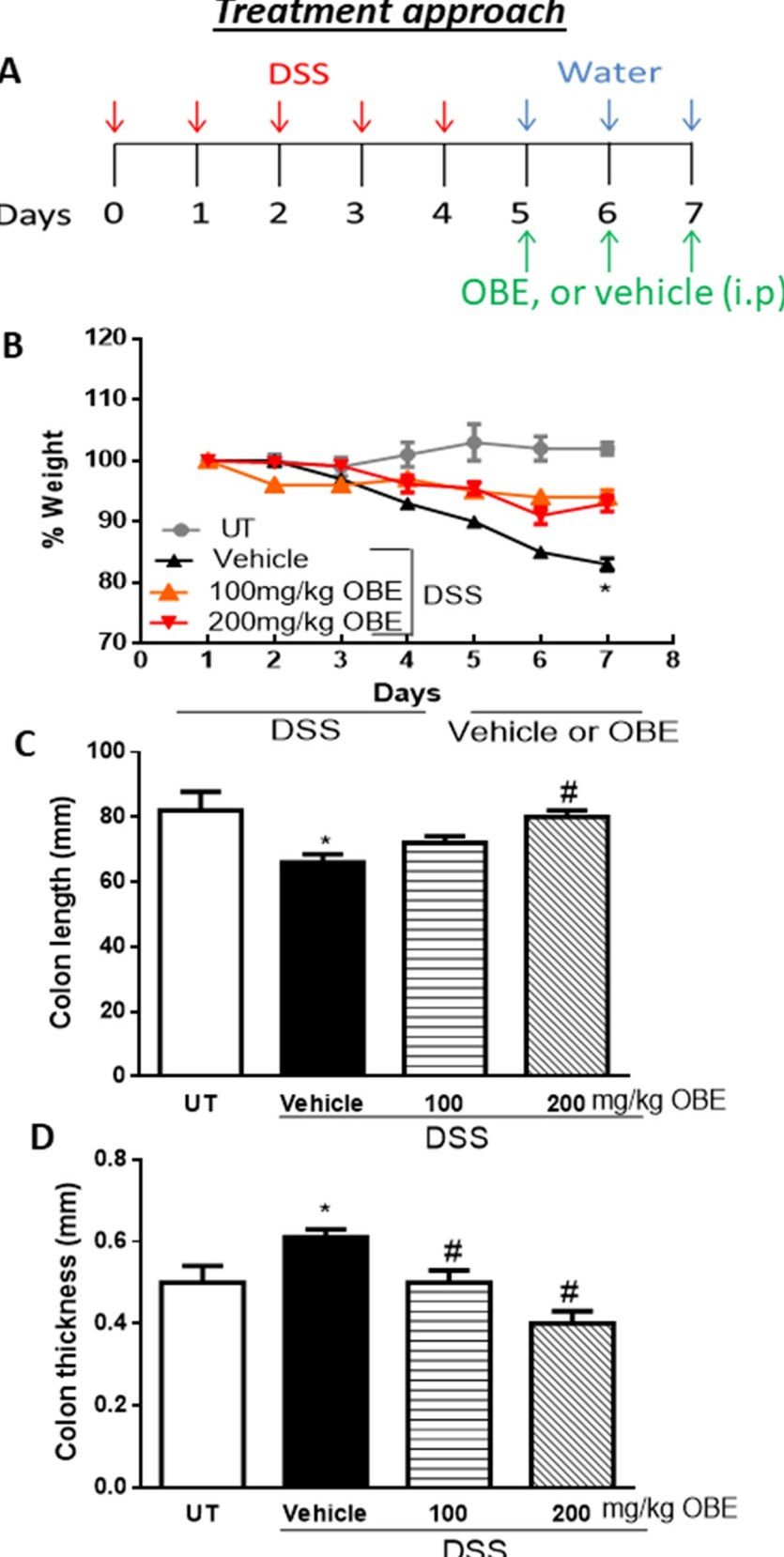

**Fig 1. Effect of OBE on colitis severity at the gross level using the treatment approach.** The treatment protocol used in the study is illustrated in panel A. Panel B shows % body weight changes in DSS-treated mice receiving vehicle (black triangles), and OBE-treated mice at various doses compared to untreated (UT) mice receiving tap water only (solid gray circles). Colon length (panel C), and thickness (panel D) were determined in mice receiving DSS and either various doses of OBE (hatched bars) or vehicle (solid bars), and in the UT healthy mice (open bars). Histobars represent means ± SEM for the following number of mice in each group: UT (n = 10), DSS/vehicle (PBS) (n = 16), DSS/OBE 100 mg/kg (n = 20), and DSS/OBE 200 mg/kg (n = 12). Asterisks denote significant difference from UT mice with p<0.05 (*), # denotes significant difference from DSS/vehicle-treated mice with p<0.05.

The gross assessment of colitis severity for this approach is shown in Table 1. Healthy (UT) mice had normal appearance, while DSS treatment resulted in diarrhea (62%), ano-rectal bleeding (43%), erythema (68%), and edema (50%). OBE treatment (100–200 mg/kg) significantly reduced the DSS-induced features of colitis (diarrhea was seen in 8–15%, ano-rectal bleeding in 5–10%, erythema in 0–15%, and edema in 0–10% of mice).

The histological assessment of colitis severity is shown in Fig 2. Mice treated with DSS plus vehicle demonstrated a significant increase in the histological score of colitis severity (10.3 ± 0.3, Fig 2A) compared to UT mice. OBE treatment moderately but significantly reduced the histological score of colitis severity in this approach (100 mg/kg dose = 8.4 ± 0.4, and 200 mg/kg dose = 8.5 ± 0.6). Also, DSS/vehicle treatment significantly increased the percentage of ulceration in the colon compared to UT mice (Fig 2B), and OBE treatment significantly reduced the percentage of ulceration in the colon by 15%. The histological features of the resected tissues are illustrated in Fig 2C. DSS/vehicle treatment resulted in significant destruction of the mucosal architecture, submucosal edema formation, increased muscle thickness, and infiltration by immune cells, and OBE treatment slightly improved mucosal integrity and reduced the degree of submucosal edema formation and immune cells recruitment to the colon.

## Effect of OBE on the colonic expression and phosphorylated levels of pro-inflammatory molecules and signaling pathways using the treatment approach

Figs 3 and 4 show IF analysis in sections from resected colon tissue of untreated (UT) mice or mice treated with DSS/vehicle and DSS/OBE (100–200 mg/kg). The colonic expression level of COX-2 (Fig 3A), and the phosphorylated levels of AKT (Fig 3B), ERK 1/2 (Fig 4A), and p38 MAPK (Fig 4B) were all significantly enhanced by DSS treatment. However, treatment with OBE significantly reduced all measured proteins to similar levels seen in the UT group. This was also confirmed with western blotting analysis (Fig 9A).

**Table 1. Effect of OBE on the macroscopic scores for colitis severity using the treatment approach.**

| Parameter | UT | Vehicle | 100 mg/kg | 200 mg/kg |
|---|---|---|---|---|
| Diarrhea | 0 | 62 * | 15 # | 8 # |
| Blood in stool | 0 | 0 | 0 | 0 |
| Ano-rectal bleeding | 0 | 43 * | 10 # | 5 # |
| Erythema | 0 | 68 * | 15 # | 0 # |
| Edema | 0 | 50 * | 10 # | 0 # |

Colitis severity was assessed in untreated (UT) mice, and in mice treated with DSS plus either vehicle or OBE.

Numbers of mice in each group are as follow: UT (n = 10), DSS/vehicle (PBS) (n = 16), DSS/OBE 100 mg/kg (n = 20), and DSS/OBE 200 mg/kg (n = 12).

* denote significant difference from UT mice

# denote significant difference from DSS/vehicle-treated mice.

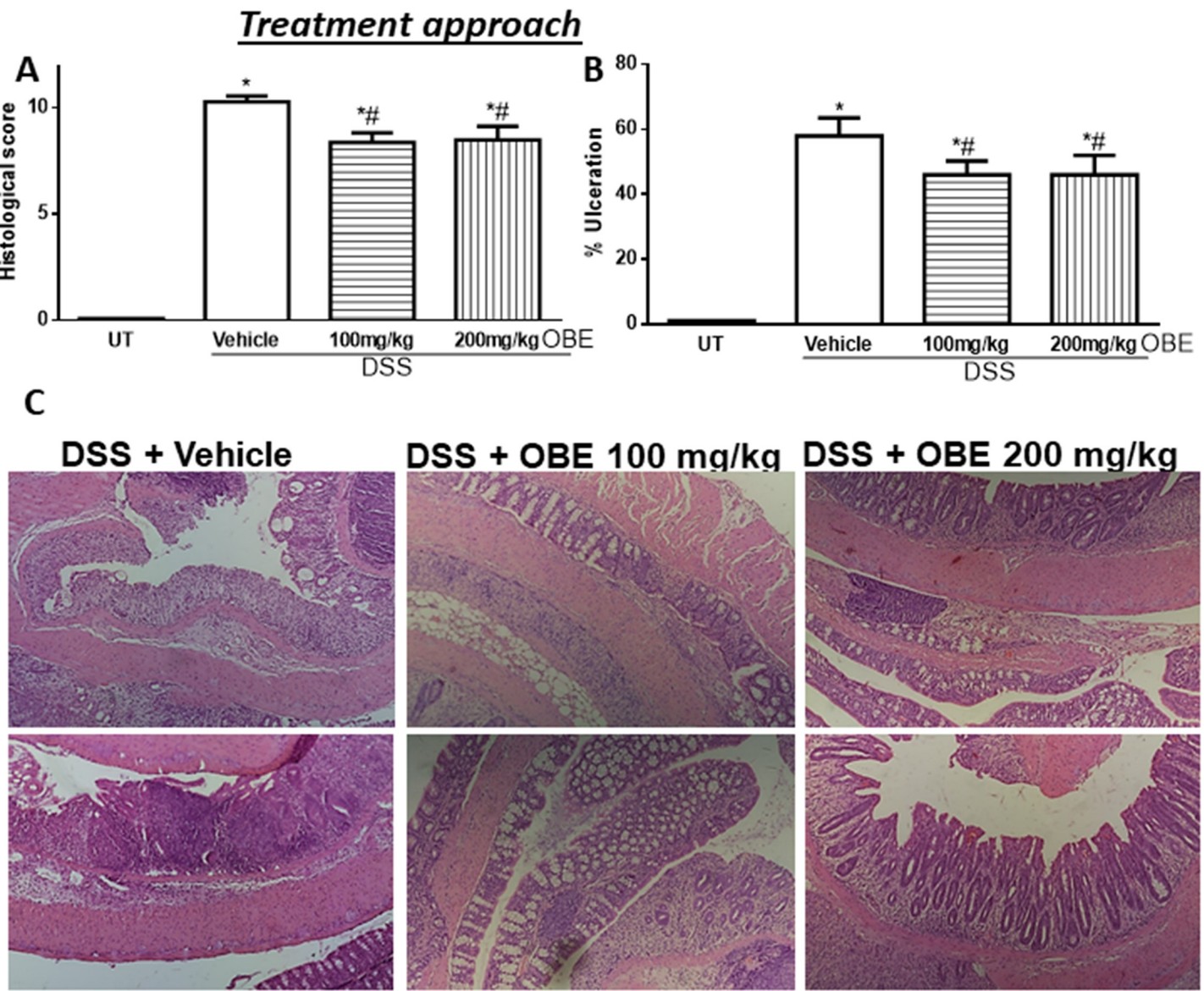

**Fig 2. Effect of OBE on colitis severity at the histological level using the treatment approach.** The histological assessment of colitis severity (panel A), and the % of ulceration in the whole colon section (panel B) were determined in mice receiving DSS and either various doses of OBE (hatched bars) or vehicle (solid bars), and in the UT healthy mice (open bars). Histobars represent means ± SEM for the following number of mice in each group: UT (n = 10), DSS/vehicle (PBS) (n = 16), DSS/OBE 100 mg/kg (n = 20), and DSS/OBE 200 mg/kg (n = 12). Asterisks denote significant difference from UT mice with $p < 0.05$ (*), # denotes significant difference from DSS/vehicle-treated mice with $p < 0.05$. Panel C (left side) is an illustration of a colon section taken from mouse treated with DSS/vehicle where there is significant mucosal destruction, submucosal edema formation, increase muscle thickness, and significant immune cells recruitment. The middle and left sides are illustrations of typical colon sections taken from mice treated with 100–200 mg/kg doses of OBE. There was a slight improvement in the mucosal integrity and slight reduction in immune cells recruitment (10x magnification).

### Effect of OBE on colitis severity at gross and histological level using the preventative approach

Using the preventative protocol described in the methodology section and in Fig 5A, control (untreated, UT) mice did not show any evidence of weight loss (Fig 5B). Mice treated with DSS + i.p/vehicle (PBS) had a significant drop in the body weight (14% from baseline level), and treatment with OBE (particularly at 200 mg/kg) partially prevented the drop in the body

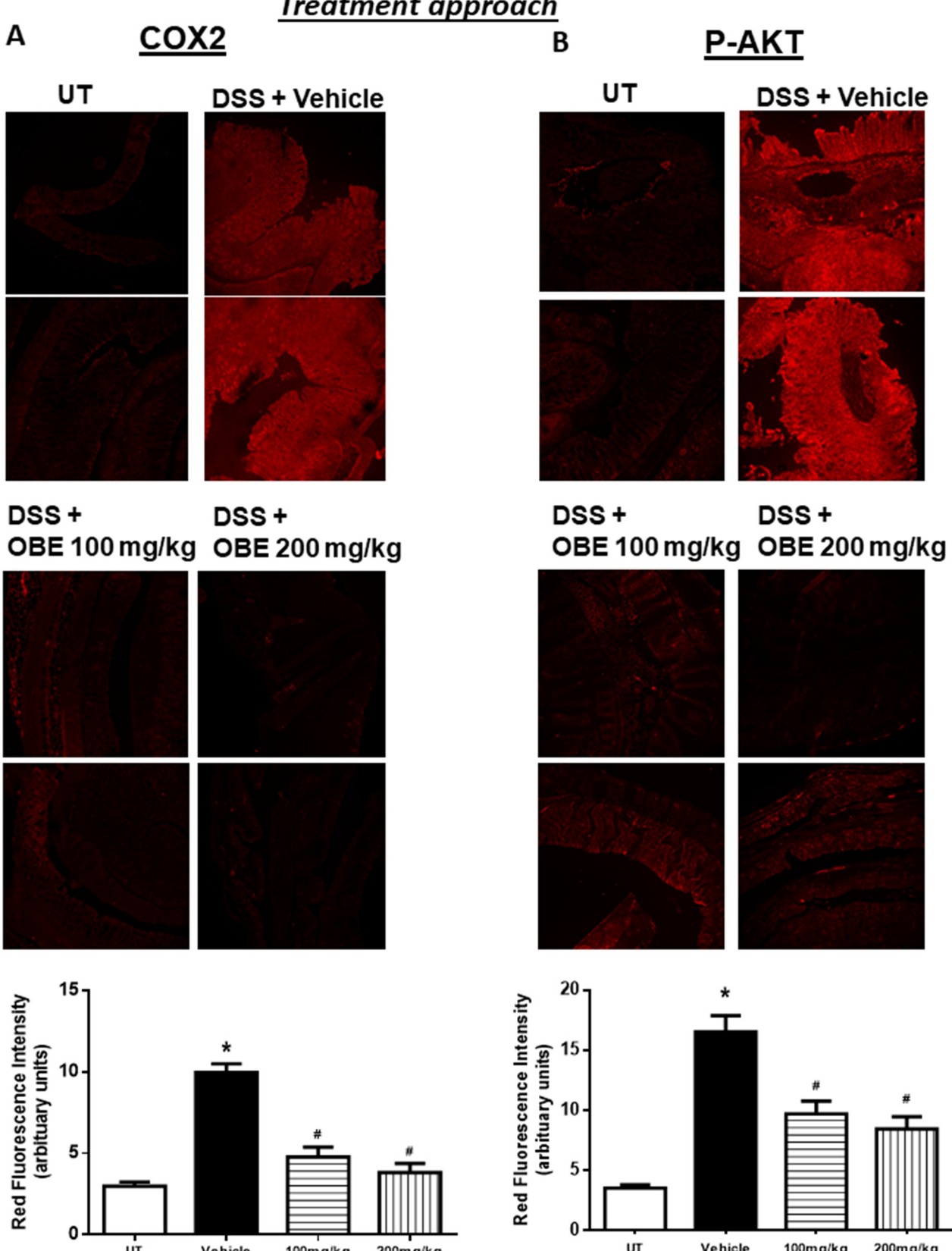

**Fig 3. Effect of OBE on colonic COX-2 expression and AKT phosphorylation levels using the treatment approach.** Colon sections taken from mice treated with DSS plus either OBE (100–200 mg/kg, hatched bars) or vehicle (solid bar) or untreated (UT, open bar) mice were immunostained with antisera against COX-2 (A), and phosphorylated AKT (B). Immunofluorescent (Alexa Fluor) signals shown in the upper side. Histobars (bottom side) represent quantitative assessment of fluorescence intensity (arbitrary units), and represent means ± SEM for 4 mice in each group. Asterisk denotes significant difference from the UT mice, with $p < 0.05$, and # denotes significant difference from DSS/vehicle-treated mice, with $p < 0.05$.

weight induced by DSS administration. In addition, the DSS-induced decrease in colon length (by 15%, Fig 5C) and the increase in colon thickness (Fig 5D) was prevented by the OBE treatment at both doses used (100–200 mg/kg).

The gross assessment of colitis severity is shown in Table 2. Healthy (UT) mice had normal appearance, while DSS treatment resulted in diarrhea (83%), blood in stool (25%), ano-rectal bleeding (83%), erythema (100%), and edema (58%). On the other hand, daily administration of OBE at 100 mg/kg dose significantly reduced the DSS-induced features of colitis, and mice treated with OBE at 200 mg/kg dose were similar to UT healthy mice with no signs of colitis at gross level.

The histological assessment of colitis severity is shown in Fig 6. Mice treated with DSS plus vehicle demonstrated a significant increase in the histological score of colitis severity (9.6 ± 0.4, Fig 6A) compared to UT mice. There was a significant reduction in the histological scores of colitis severity by OBE treatment (100 mg/kg dose = 7 ± 0.5, and 200 mg/kg dose = 5 ± 0.6). DSS/vehicle treatment also significantly increased the percentage of ulceration in the colon compared to UT mice (Fig 6B), and OBE treatment significantly reduced the percentage of ulceration in the colon by 20–60%. The histological features of the resected tissues are illustrated in Fig 6C. DSS plus vehicle treatment resulted in significant destruction of the mucosal architecture, submucosal edema formation, increased muscle thickness, and infiltration by immune cells, and OBE treatment (particularly at 200 mg/kg) improved mucosal integrity and reduced the degree of submucosal edema formation and immune cells recruitment to the colon.

## Effect of OBE on the colonic expression and phosphorylated levels of pro-inflammatory molecules and signaling pathways using the preventative approach

Figs 7 and 8 show immunofluorescence (IF) analysis in sections from resected colon tissue of untreated (UT) mice or mice treated with DSS/vehicle and DSS/OBE (100–200 mg/kg). The colonic expression level of COX-2 (Fig 7A), and the phosphorylated levels of AKT (Fig 7B), ERK 1/2 (Fig 8A), and p38 MAPK (Fig 8B) were all significantly enhanced by DSS treatment. In contrast treatment with OBE significantly reduced all proteins to roughly similar levels seen in the UT group. This was also confirmed with western blotting analysis (Fig 9A).

## Effect of OBE on the colonic expression of various immune cell markers

Using western blotting analysis, DSS treatment enhanced the expression level of neutrophil (LY6G) (but not macrophage, F4/80) marker, and this was reduced by OBE treatment using the preventative and treatment approaches (Fig 9A). There was no expression profile seen for T-, and B-cell markers in the colon of UT or DSS+\-OBE treated mice even when very high concentration of primary and secondary antibodies were used (data not shown).

## Effect of OBE on neutrophil superoxide release and survival *in vitro*

Treatment of freshly isolated bone-marrow derived neutrophils from naïve mice with the bacterially-derived stimulus fMLP-like peptide WKYMVm (10 μM) significantly enhanced superoxide release (Fig 9B). OBE treatment significantly decreased the WKYMVm-induced

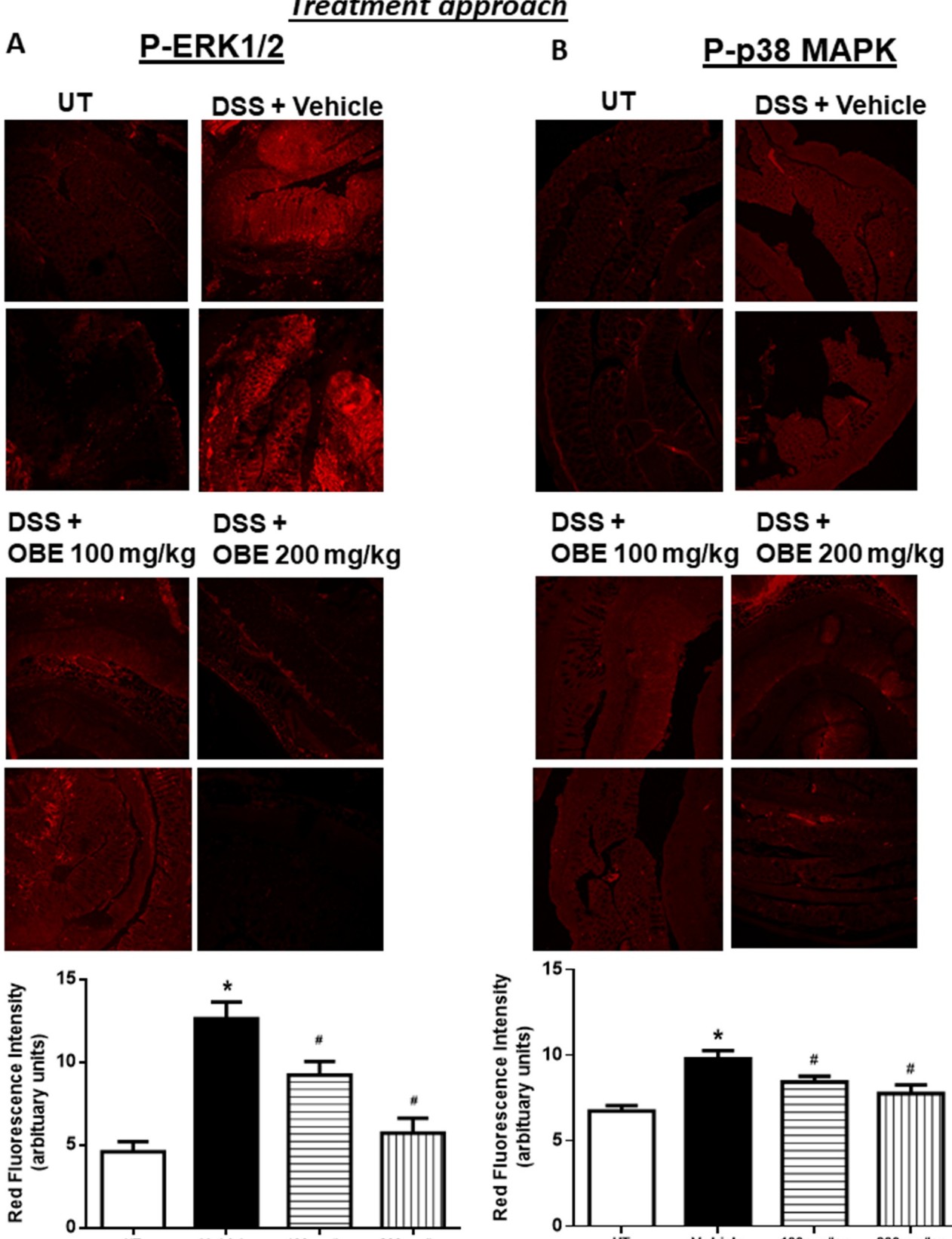

**Fig 4. Effect of OBE on colonic phosphorylation levels of ERK 1/2 and p38 MAPK using the treatment approach.** Colon sections taken from mice treated with DSS plus either OBE (100–200 mg/kg, hatched bar) or vehicle (solid bar) or untreated (UT, open bar) mice were immunostained with antisera against phosphorylated ERK 1/2 (A) or p38 MAPK (B). Immunofluorescent (Alexa Fluor) signals shown in upper side. Histobars (bottom side) represent quantitative assessment of fluorescence intensity (arbitrary units), and represent means ± SEM for 4 mice in each group. Asterisk denotes significant difference from the UT mice, with $p < 0.05$, and # denotes significant difference from DSS/vehicle-treated mice, with $p < 0.05$.

superoxide release in a dose-dependent manner and was completely abolished at 10–100 μg/ml concentrations of OBE. Superoxide release was also inhibited by superoxide dismutase treatment (positive control, data not shown).

In addition, OBE treatment (1–100 μg/ml) significantly reduced the percentage of viable bone-marrow derived neutrophils (Fig 9C) and increased their spontaneous apoptosis (Fig 9D) relative to vehicle (PBS) treated cells.

## Discussion

Most of the documented anti-inflammatory activities shown by onion are attributed to the known flavonoid quercetin, where polar solvents, particularly water and/or alcohols, were used to prepare the tested extracts. Therefore, in order to investigate the possibility of having other ingredients with promising activities, particularly in reducing the colitis severity, the applied extraction method in this study was selected to provide extracts with, mainly, non-polar constituents, which, in a way, resembles the onion essential oils, where the highly polar bioactive flavonoids, including quercetin, are not among them. This conclusion was further confirmed through the chromatographic analysis of the extracts. Thus, dichloromethane was selected, as the extraction solvent, over water or alcohols. Hence, the observed promising activities of the extract in reducing the colitis severity can be attributed to the non-polar constituents. These fat-soluble constituents are likely to be the sulfur-containing compounds which are found excessively in the essential oil of onion.

In this study, we tried to answer two important questions; a) can OBE treatment prevent/reduce features of active state of colitis, and b) if OBE given prophylactically, will it prevent/reduce colitis severity. Our findings show that OBE administration either before (preventative approach) or after (therapeutic treatment approach) colitis induction significantly reduced the degree of the inflammatory response at both gross and histological levels. This is in agreement with our recent study showing anti-inflammatory properties of OBE when was given at a similar time frame as the colitis inducing agent, DSS [25]. These findings therefore clearly confirm that OBE may have beneficial effects to patients with active colitis and can also reduce the probability of developing colitis in patients at risk.

All of the IBD patients are treated (with anti-inflammatory, immunosuppressants, or biological therapies) once the colitis is established, and it was important to test if OBE can reduce colitis severity in mice once it is established. As shown in Figs 1 and 2 and Table 1, using the therapeutic treatment approach, OBE (100–200 mg/kg) moderately (at microscopic level) but significantly reduced colitis severity in mice. In comparison with our previously published report using OBE administration along colitis induction (prophylactic approach) [25], the prophylactic approach reduced all gross parameters of colitis severity by 96–100%, which is comparable to the therapeutic treatment approach used in this study (gross parameters were reduced by 80–100%, Table 1). However, at the histological level, OBE was most potent in reducing colitis severity in mice when administered along with colitis induction using the prophylactic approach [25] with approximately 50–60% reduction in the histological score for colitis at 100–200 mg/kg doses respectively, whereas in the therapeutic treatment approach, OBE marginally (10% at 100–200 mg/kg doses) but significantly reduced the histological score of colitis (Fig 2A). A possible explanation for this is that it is easier to reduce the severity of

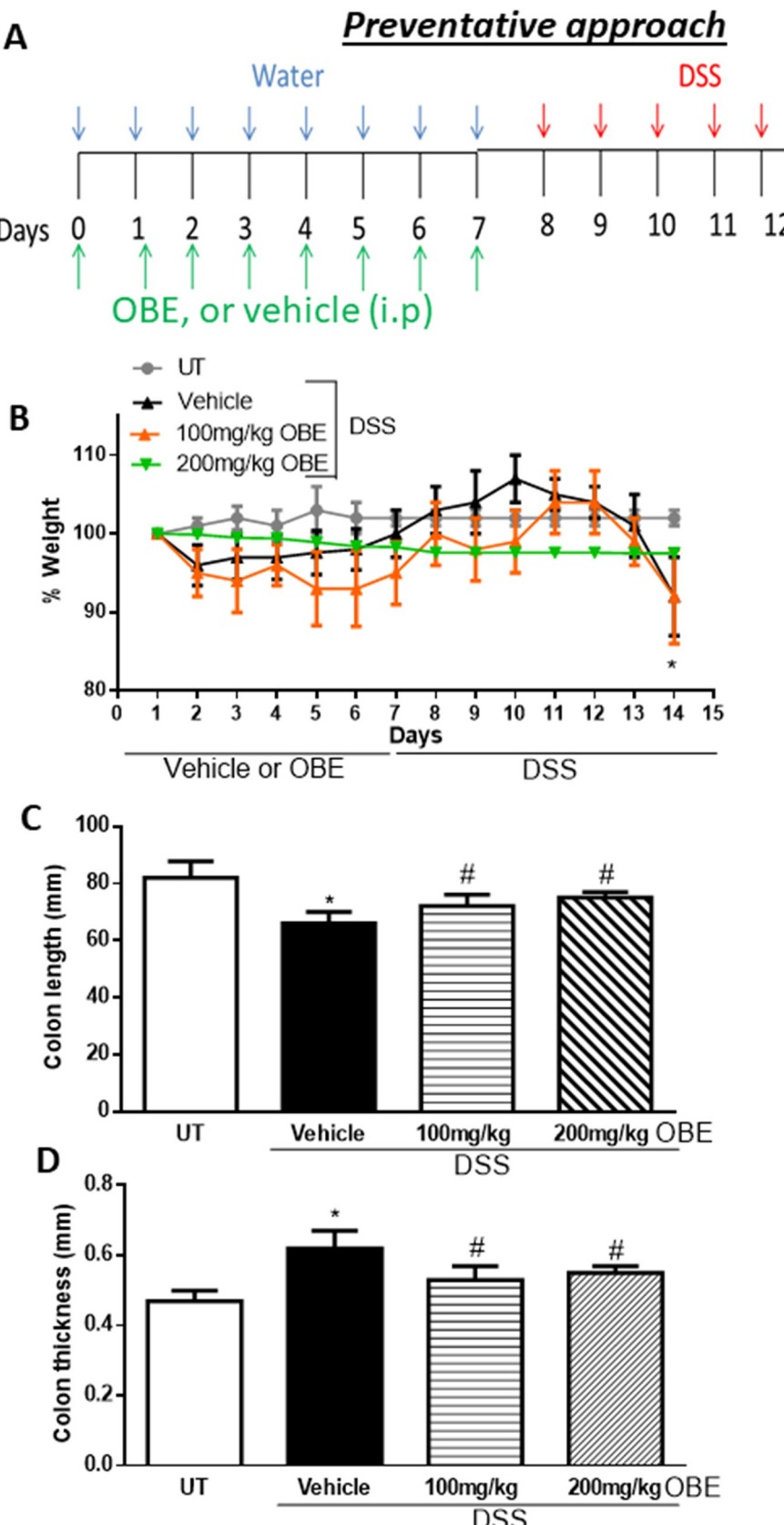

**Fig 5. Effect of OBE on colitis severity at the gross level using the preventative approach.** The treatment protocol used in the study is illustrated in panel A. Panel B shows % body weight changes in DSS-treated mice receiving vehicle (black triangles), and OBE-treated mice at various doses compared to untreated (UT) mice receiving tap water only (solid gray circles). Colon length (panel C), and thickness (panel D) were determined in mice receiving DSS and either vehicle (solid bars), or various doses of OBE (hatched bars) and in the UT mice (open bars). Histobars represent means ± SEM for the following number of mice in each group: UT (n = 7), DSS/vehicle (PBS) (n = 12), DSS/OBE 100 mg/kg (n = 13), and DSS/OBE 200 mg/kg (n = 7). Asterisks denote significant difference from UT mice with p<0.05 (*), # denotes significant difference from DSS/vehicle-treated mice with p<0.05.

colitis when the treatment is started at early stage rather than once it is fully established. In regards of using the preventative approach, our data here (Figs 5 and 6 and Table 2) show that this approach works. At gross levels, the anti-inflammatory effect of preventative approach was comparable to the prophylactic and therapeutic treatment approaches. At the histological level, the anti-inflammatory effect of OBE using the preventative approach was better compared to the therapeutic treatment approach with around 30–50% reduction in the histological score of colitis at 100–200 mg/kg doses (Fig 6A). Our data suggest that OBE is more beneficial in reducing colitis severity when given as a dietary approach in patients with high risk and in patients at very early stage of diagnosis, but its effect is minimal when given for patients with chronic/active state of colitis.

Several studies have also shown beneficial effects of regular onion consumptions in the prevention and/or treatment of several disease conditions in pre-clinical and clinical settings. For example, regular onion powder consumption (7% w/w for 7 weeks) in rats prevented the development of non-alcoholic fatty liver disease and its beneficial effects seemed to be, at least in part, mediated through reduction of the serum levels of alanine aminotransferase, triglycerides, and glucose, hepatic TNFα levels, and lower hepatic stenosis and lobular and portal inflammation [17]. Furthermore, another study reported that consumption of red wine extract of onion juice, by hypercholesterolemic individuals (250 ml daily for 10 weeks) significantly reduced total cholesterol and low density lipoprotein levels, and the inflammatory marker factor VII suggesting a cardio-protective roles for onion in individuals with risk factors for the development of cardiovascular diseases [18]. A randomized double-blinded placebo-controlled cross-over trial involving 70 patients suggested that supplementation of quercetin (162 mg per day for 6 weeks) derived from onion skin extracts significantly reduced systolic blood pressure (by 3.6 mmHg) in hypertensive patients also suggesting cardio-protective roles for quercetin [19]. Also, food frequency questionnaire data in China suggested that high consumption of onion, garlic, and shallots inversely associated with the risk of development gallbladder cancer [20]. With regards to colitis, several dietary approaches have been investigated.

**Table 2. Effect of OBE on the macroscopic scores for colitis severity using the preventative approach.**

| Parameter | UT | Vehicle | 100 mg/kg | 200 mg/kg |
|---|---|---|---|---|
| Diarrhea | 0 | 83 * | 69 * | 0 # |
| Blood in stool | 0 | 25 * | 15 * | 0 # |
| Ano-rectal bleeding | 0 | 83 * | 23 *# | 0 # |
| Erythema | 0 | 100 * | 46 *# | 0 # |
| Edema | 0 | 58 * | 61 * | 0 # |

Colitis severity was assessed in untreated (UT) mice, and in mice treated with DSS plus either vehicle or OBE.
Numbers of mice in each group are as follow: UT (n = 7), DSS/vehicle (PBS) (n = 12), DSS/OBE 100 mg/kg (n = 13), and DSS/OBE 200 mg/kg (n = 7).
* denote significant difference from UT mice
# denote significant difference from DSS/vehicle-treated mice.

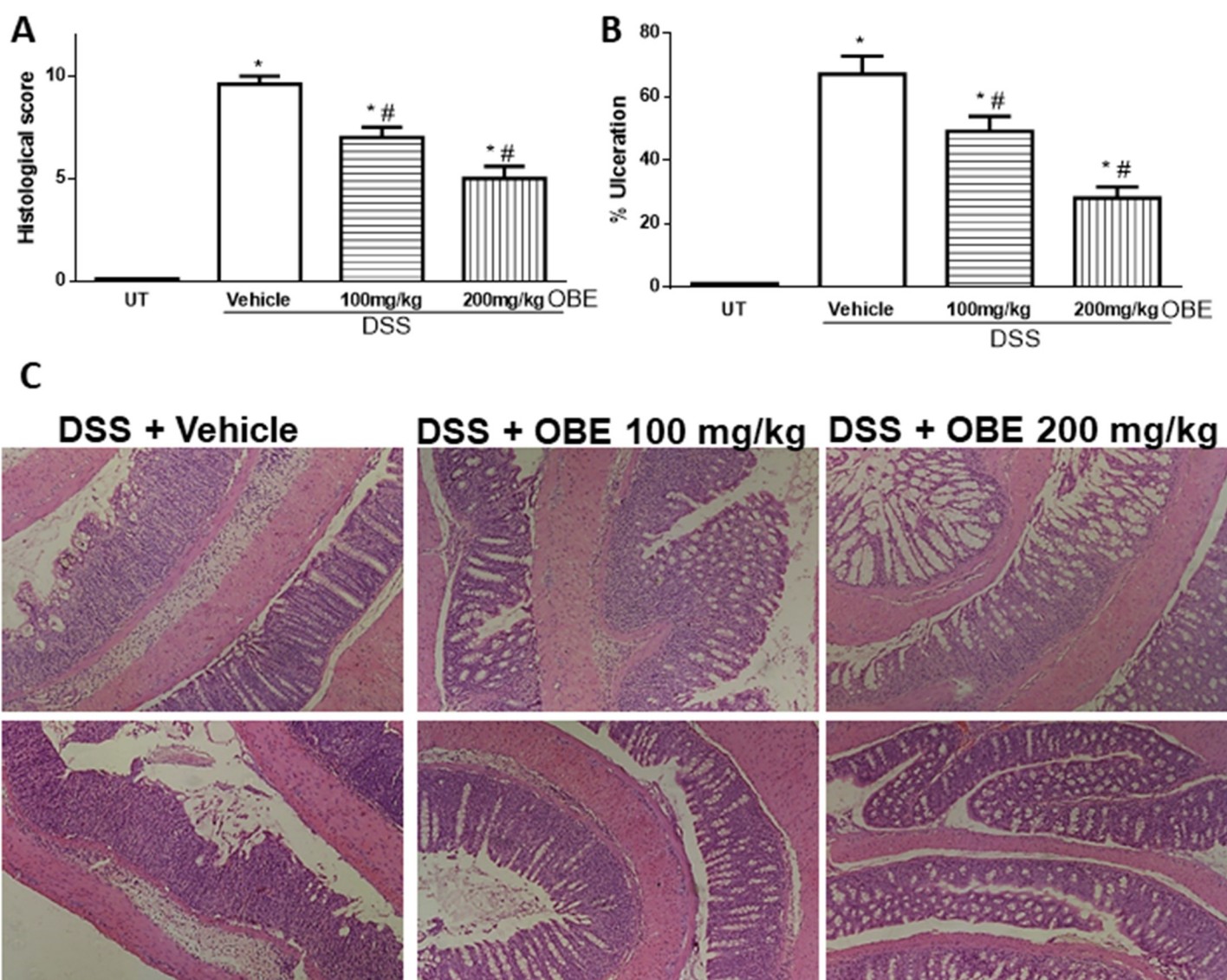

**Fig 6. Effect of OBE on colitis severity at the histological level using the preventative approach.** The histological assessment of colitis severity (panel A), and the % of ulceration in the whole colon section (panel B) were determined in mice receiving DSS and either vehicle (solid bars), or various doses of OBE (hatched bars) and in the UT healthy mice (open bars). Histobars represent means ± SEM for the following number of mice in each group: UT (n = 7), DSS/vehicle (PBS) (n = 12), DSS/OBE 100 mg/kg (n = 13), and DSS/OBE 200 mg/kg (n = 7). Asterisks denote significant difference from UT mice with p<0.05 (*), # denotes significant difference from DSS/vehicle-treated mice with p<0.05. Panel C (left side) is an illustration of a colon section taken from mouse treated with DSS/vehicle where there is significant mucosal destruction, submucosal edema formation, increase muscle thickness, and significant immune cells recruitment. The middle and left sides are illustrations of typical colon sections taken from mice treated with 100–200 mg/kg doses of OBE. There was an improvement in the mucosal integrity and reduced the enhanced immune cells recruitment (10x magnification).

The results have been mixed with some studies showing beneficial effects, in part through modulating the composition of the gut microflora and the expression/activity of pro-inflammatory molecules, whilst others have reported deleterious effects. For example, soft drink consumption and sucrose or red meat intake were associated with increased risk of ulcerative colitis (UC) development [34–36]. In addition, western type-, high salt-type, and carrageenan-free-type of diet was linked with increased colitis severity in mice [37, 38] and in IBD patients

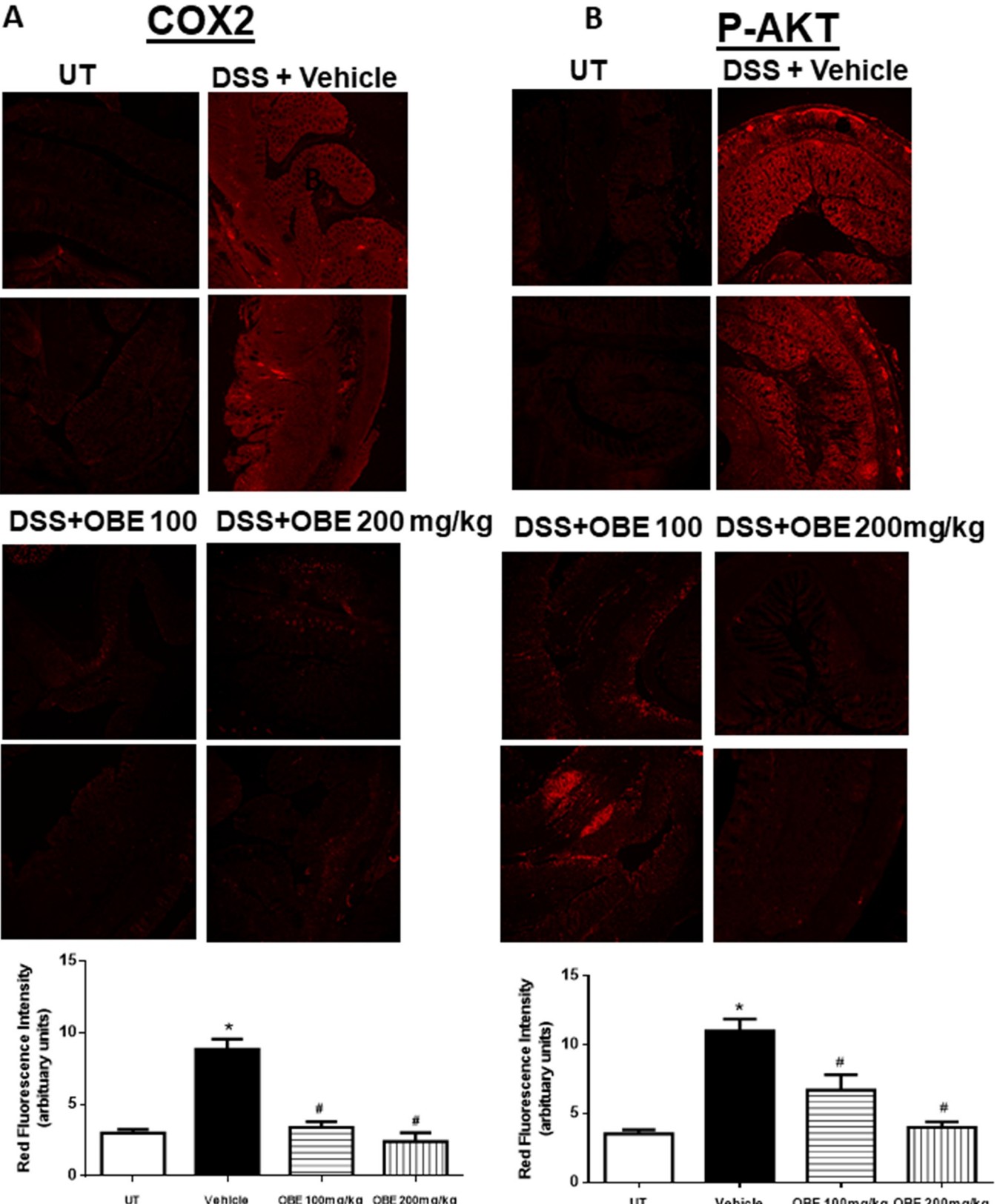

**Fig 7. Effect of OBE on colonic COX-2 expression and AKT phosphorylation levels using the preventative approach.** Colon sections taken from mice treated with DSS plus either vehicle (solid bar) or OBE (100–200 mg/kg, hatched bar) or untreated (UT, open bar) mice were immunostained with antisera against COX-2 (A), and phosphorylated AKT (B). Immunofluorescent (Alexa Fluor) signals shown in left side. Histobars (right side) represent quantitative assessment of fluorescence intensity (arbitrary units), and represent means ± SEM for 4 mice in each group. Asterisk denotes significant difference from the UT mice, with p<0.05, and # denotes significant difference from DSS/vehicle-treated mice, with p<0.05.

[39, 40]. Zinc deficiency was also linked with increased intestinal barrier permeability and colitis exacerbation in IBD patients [41]. On the other hand, consumption of diet high in fruits and vegetables was associated with reduced colitis severity [42–44]. Also, the monosaccharide fructose administration in mice reduced colitis severity through modulating the gut microbiota [45]. Our novel data presented here provides a rational basis for using onion as a dietary preventative measure for patients at risk for IBD development.

Various pro-inflammatory signaling molecules were shown to play a role in colitis pathogenesis. For example, enhanced expression/activity profile of the MAPK family members (p38 MAPK, JNK, and ERK1/2) [27, 46], PI3K [47, 48], and the pro-inflammatory COX-2 [49–52] was observed in the colonic biopsies of IBD patients as well as in different animal models of colitis [25]. In addition, using chemical inhibitors of these pro-inflammatory molecules ameliorated colitis severity. A recent study demonstrated an important role for COX-2 in colitis through enhancing the expression profile of Th17 cytokines [53]. Another study showed enhanced COX-2 colonic expression in TNBS-treated mice and quercitin reduced colitis severity through decreasing COX-2 expression [54]. In our study, DSS administration enhanced the colonic expression profile of COX2, and the phosphorylated levels of p38 MAPK, ERK1/2 and Akt; this was significantly reduced with OBE (to baseline levels seen in UT mice) by using the preventative and therapeutic treatment approaches (Figs 3, 4 and 7–9A). These data are also in agreement with the anti-inflammatory effects of OBE in the DSS model using the prophylactic approach [25], suggesting that OBE utilizes similar downstream molecules in its anti-inflammatory effects irrespective of the approach used. Furthermore, having a natural product able to target multiple arms of the inflammatory signaling cascade is of significant importance since targeting one molecule might not be satisfactory to control the complex inflammatory melui seen in colitis.

Polymorphnuclear cells (or neutrophils) are the most abundant type of circulating white blood cells and play an essential role in IBD pathogenesis. Enhanced degree of neutrophil recruitment to the colon was observed in animal models of colitis [31, 55, 56] and in IBD patients [57]. High levels of neutrophil extracellular traps (NETs) were detected in the plasma and colon tissue of IBD patients with active disease state as well as in mice treated with DSS. Inhibition of NET release (through DNase administration) ameliorated colitis as well as colitis-associated tumorigenesis [58]. In addition, neutrophils are one of the major source of reactive oxygen species (ROS) which trigger tissue injury and lead to chronic inflammation and tumorigenesis [59, 60]. Reducing the degree of ROS by various means was shown to reduce colitis severity [61–66]. In our study, we aimed to extend our understanding of the anti-inflammatory properties of OBE (irrespective of the various therapeutic approaches used *in vivo*), and we showed that OBE treatment significantly reduced the degree of neutrophil recruitment to the colon (Fig 9A) and superoxide release from bone-marrow derived neutrophils in response to a bacterial stimulus in a dose-dependent manner (Fig 9B), and enhanced their spontaneous apoptosis (Fig 9C and 9D).

## Conclusions

In conclusion, we showed that OBE treatment can reduce colitis severity in mice when administered either before or after colitis induction, in part through reducing the expression/activity

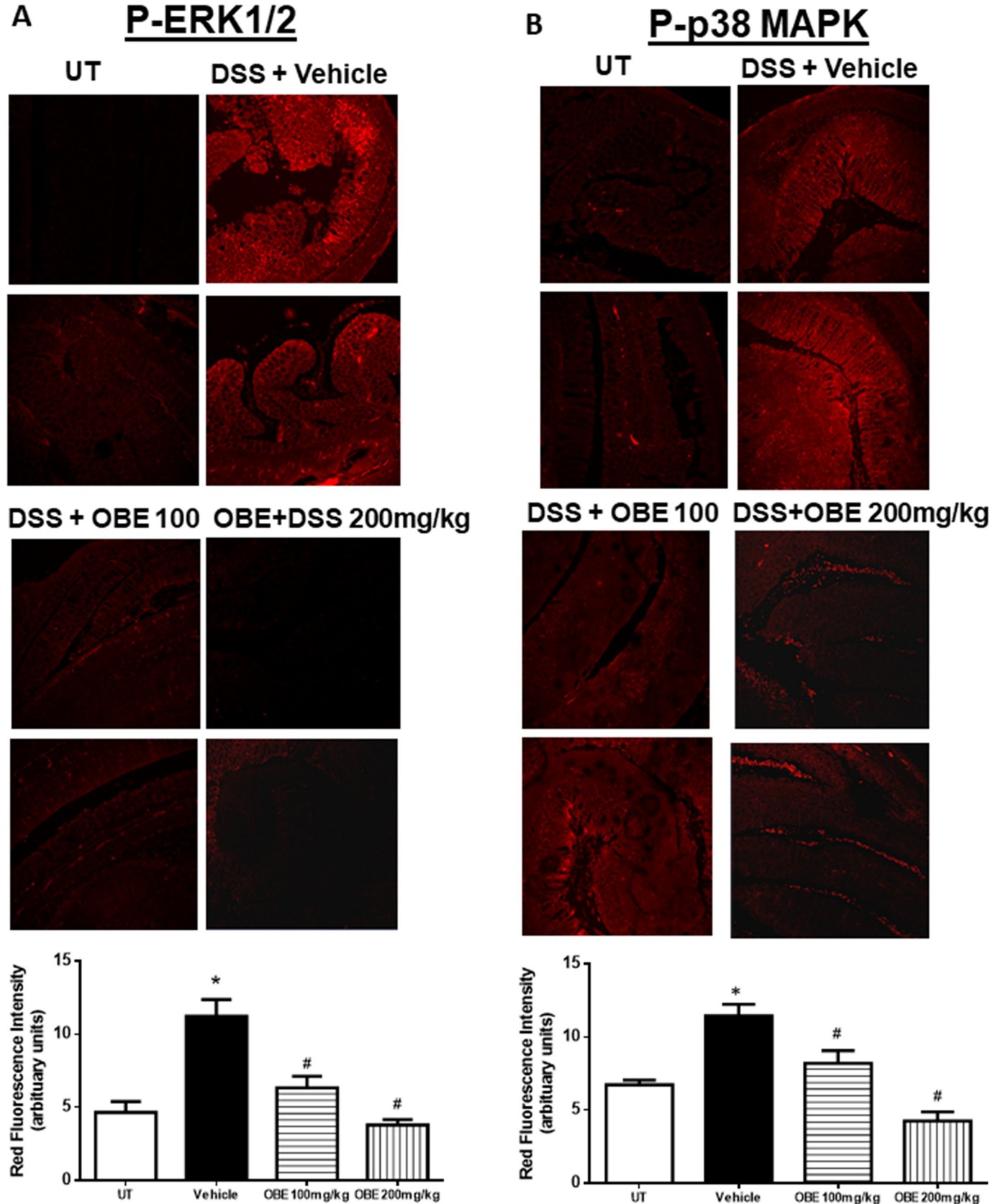

**Fig 8. Effect of OBE on colonic phosphorylation levels of ERK 1/2 and p38 MAPK using the preventative approach.** Colon sections taken from mice treated with DSS plus either vehicle (solid bar) or OBE (100–200 mg/kg, hatched bar) or untreated (UT, open bar) mice were immunostained with antisera against phosphorylated ERK 1/2 (A) or p38 MAPK (B). Immunofluorescent (Alexa Fluor) signals shown in left side. Histobars (right side) represent quantitative assessment of fluorescence intensity (arbitrary units), and represent means ± SEM for 4 mice in each group. Asterisk denotes significant difference from the UT mice, with p<0.05, and # denotes significant difference from DSS/vehicle-treated mice, with p<0.05.

of important pro-inflammatory molecules. Based on the degree of reduction in the histological score of colitis by OBE using various therapeutic approaches, we recommend using onion (OBE) as a dietary prophylactic approach in patients with high risk and in a recently diagnosed

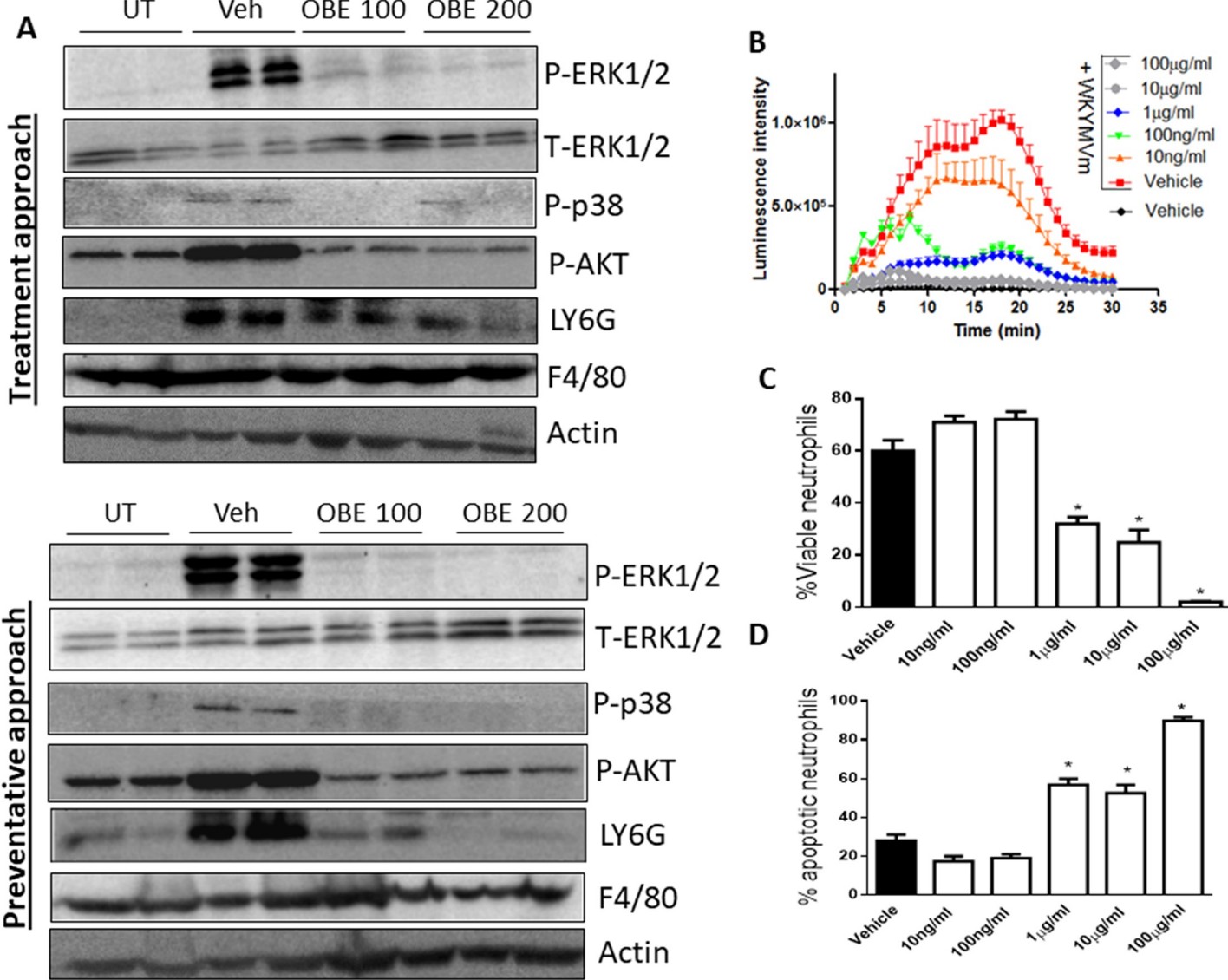

**Fig 9. Effect of OBE on the expression/activity of pro-inflammatory molecules and immune cell markers, and neutrophil superoxide release and spontaneous apoptosis *in vitro*.** Colon sections taken from mice treated for with DSS/vehicle, DSS/OBE (100–200 mg/kg) or untreated (UT) mice were immunostained with antisera against phosphorylated/total ERK1/2, AKT, p38, LY6G, F4/80, or actin (Panel A). The blots represent one of 3 similar experiments. Neutrophils were isolated from naïve mice and the level of superoxide release was determined by recording chemiluminescence emanating from oxidation of luminol substrate. Neutrophils were treated with vehicle only (solid circles), or 10 μM WKYMVm plus vehicle (red squares) or various concentrations of OBE (panel B). The % of viable (panel C) and % apoptotic (panel D) neutrophils in response to vehicle (open bar) or various concentrations of OBE (hatched bars) was determined using Annexin-V/7AAD assay. Histobars represent means ± SEMs for 3 independent experiments conducted with cells isolated from 3 mice in each group. Asterisks denote significant difference from vehicle group, with p<0.05.

patients with mild colitis. Its efficacy in reducing an active state of chronic colitis might be marginal when given alone. We also provided novel *in vitro* data showing a direct role for OBE in reducing neutrophil survival and ROS release.

## Supporting information

**S1 Fig.**
(TIF)

**S1 Raw data.**
(DOCX)

## Acknowledgments

We would like to thank Dr. Ananthalakshmi KV for the technical help, the histology unit in the faculty of Allied Health, and the animal unit in Health Science Center, Kuwait University.

## Author Contributions

**Conceptualization:** Maitham A. Khajah.

**Data curation:** Maitham A. Khajah.

**Formal analysis:** Maitham A. Khajah.

**Funding acquisition:** Maitham A. Khajah, Ahmed Z. EL-Hashim, Khaled Y. Orabi.

**Investigation:** Maitham A. Khajah.

**Methodology:** Maitham A. Khajah, Khaled Y. Orabi, Sanaa Hawai, Hanan G. Sary.

**Project administration:** Maitham A. Khajah, Ahmed Z. EL-Hashim.

**Resources:** Maitham A. Khajah, Ahmed Z. EL-Hashim, Khaled Y. Orabi.

**Supervision:** Maitham A. Khajah, Khaled Y. Orabi.

**Validation:** Maitham A. Khajah, Ahmed Z. EL-Hashim.

**Writing – original draft:** Maitham A. Khajah.

**Writing – review & editing:** Ahmed Z. EL-Hashim, Khaled Y. Orabi.

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
