## [Decision Letter · Decision Letter 0]

22 Jul 2020

PONE-D-20-14395

Onion bulb extract can both reverse and prevent colitis in mice via inhibition of pro-inflammatory signaling molecules and neutrophil activity

PLOS ONE

Dear Dr. Khajah,

Thank you for submitting your manuscript to PLOS ONE. After careful consideration, we feel that it has merit but does not fully meet PLOS ONE’s publication criteria as it currently stands. Therefore, we invite you to submit a revised version of the manuscript that addresses the points raised during the review process.

Both reviewers think that the conclusions are partly supported by the data. So I would recommend to pay attention to strengthen the results and to avoid overstating the conclusions.

Please submit your revised manuscript within 3 months. If you will need more time than this to complete your revisions, please reply to this message or contact the journal office at plosone@plos.org. Please include the following items when submitting your revised manuscript:

We look forward to receiving your revised manuscript.

Kind regards,

Mathilde Body-Malapel

Academic Editor

PLOS ONE

Journal Requirements:

2. To comply with PLOS ONE submissions requirements, in your Methods section, please provide additional information on the animal research and ensure you have included details on (a) methods of sacrifice, (b) methods of anesthesia and/or analgesia, and (c) efforts to alleviate suffering.

4. Please include your tables as part of your main manuscript and remove the individual files.

Please note that supplementary tables should be uploaded as separate "supporting information" files.

Reviewers' comments:

Reviewer's Responses to Questions

**Comments to the Author**

1. Is the manuscript technically sound, and do the data support the conclusions?

Reviewer #1: Partly

Reviewer #2: Partly

2. Has the statistical analysis been performed appropriately and rigorously? 

Reviewer #1: Yes

Reviewer #2: I Don't Know

3. Have the authors made all data underlying the findings in their manuscript fully available?

Reviewer #1: Yes

Reviewer #2: Yes

4. Is the manuscript presented in an intelligible fashion and written in standard English?

Reviewer #1: Yes

Reviewer #2: Yes

5. Review Comments to the Author

Reviewer #1: In the current study, authors investigated the impact of Onion bulb extract (OBE) on colitis severity in BALB/c mice models. Here are my concerns:

1- The methodology was not well described; the extraction procedure, macroscopic assessment of colitis severity, immunofluorescence, and isolation of bone-marrow derived neutrophils methods should be well explained.

2- Estimation of the used OBE doses should be explained.

3- Why they used IP injection instead of oral administration?

4- It is not cleared they used an alcoholic OBE extract or aqueous OBE extract.

5- Herbs extracts contain various active ingredients like flavonoids and etc., the authors should be specified that which compound is responsible for the observed modulatory results in their investigation.

6- The results of OBE 200 mg/kg impact on pro-inflammatory molecules are not shown in the Figures 3 & 4 for preventative approach.

7- Sample size estimation is not clarified. The number of mice are varied between examined groups and may impact on the overall conclusion. Why we had more animal subjects (almost double) in 100 OBE mg/kg group than the 200 OBE mg/kg group?

8- For a better conclusion, Comparative analysis between two different OPE doses in all of the examinations is needed.

Reviewer #2: The manuscript by Dr. Maitham Khajah et al., suggests that Onion bulb extract (OBE) can reverse and prevent colitis in mice via inhibition of pro-inflammatory signaling molecules and neutrophil activity. The authors demonstrated that OBE treatment significantly reduced colitis severity, the colonic expression/activity profile of pro-inflammatory molecules, inhibited neutrophils-induced superoxide release, and increased spontaneous apoptosis of neutrophils in vitro. The data demonstrated the anti-inflammatory properties of onion bulb extract (OBE) in reducing colitis severity in mice. The study is well designed and logical but partially supported by the data. The experiments are well developed, need some in-vivo effector site protection of colon as compared to bone marrow-derived induced neutrophils function. The specific comments are:

1. The figure shows the effects of OBE on DSS induced colitis. The changes in colon infiltrating cells is crucial for any effector function is missing. Further, it is not clear that what is the maximum effective dose as 100 or 200 mg/kg body weight.

2. The percentage of changes in body weight data is very nice. It is not clear as to which immune cells recruited to the colon and OBE treatment reduce/enhances those immune populations.

3. Need a strong rationale for using bone marrow-derived neutrophils in place of neutrophils present in the colon during DSS and OBE treatment.

4. Histological data at both doses ( 100 or 200 mg/kb ) shows similar results. I suggest PI to just follow one dose for clarity in this manuscript.

5. The observation of changes in COX-2, P-AKT, P ERK1/2 through immunofluorescence analysis is not clear and needs a better technique like western for a prudent conclusion.

6. Fig.7 is a very poor quality presentation. It is not clear as cells are making these or a just antibodies background. Need some better techniques like immunocytochemistry ( works better in paraffin section).

Taken together, although the study is interesting and meaningful, but need mechanistic supporting data from colon site/immune cells for a prudent conclusion.

6. PLOS authors have the option to publish the peer review history of their article (what does this mean?). If published, this will include your full peer review and any attached files.

Reviewer #1: **Yes: **Ehsan Gharib

Reviewer #2: No

---

## [Author Response · Author response to Decision Letter 0]

29 Sep 2020

Response to reviewers’ comments 

First, we would like to thank the reviewers for their valuable comments to increase the quality of this manuscript. 

Reviewer #1: 

In the current study, authors investigated the impact of Onion bulb extract (OBE) on colitis severity in BALB/c mice models. Here are my concerns:

1- The methodology was not well described; the extraction procedure, macroscopic assessment of colitis severity, immunofluorescence, and isolation of bone-marrow derived neutrophils methods should be well explained.

Response: a detailed methodology for the above mentioned procedures is now included in the revised version of the manuscript. We also performed western blotting analysis for the expression/activity of various signaling molecules and immune cell markers as suggested by the reviewers, and therefore we included a methodology section for this technique in the revised version of the manuscript. 

2- Estimation of the used OBE doses should be explained.

Response: the following sentence is now included in the (result) section: 

We choose 100-200 mg/kg dose of OBE based on our previously published report using the prophylactic approach [25] which demonstrated anti-inflammatory properties using the same colitis model. 

3- Why they used IP injection instead of oral administration?

Response: we tried oral gavage and coating the food with OBE but we did not observe any anti-inflammatory properties of OBE (unpublished data). Therefore, we used the i.p route and clearly showed anti-inflammatory properties using various approaches; prophylactic (previous publication, Ref 25), and preventative and treatment (current manuscript).

4- It is not cleared they used an alcoholic OBE extract or aqueous OBE extract.

Response: dichloromethane was used as the extraction solvent. This is now clear after adding the detailed extraction procedure to the revised version of the manuscript.

The following paragraph is now included in the discussion section of the revised version of the manuscript: 

Most of the documented anti-inflammatory activities shown by onion are attributed to the known flavonoid quercetin, where polar solvents, particularly water and/or alcohols, were used to prepare the tested extracts. Therefore, in order to investigate the possibility of having other ingredients with promising activities, particularly in reducing the colitis severity, the applied extraction method in this study was selected to provide extracts with, mainly, non-polar constituents, which, in a way, resembles the onion essential oils, where the highly polar bioactive flavonoids, including quercetin, are not among them. This conclusion was further confirmed through the chromatographic analysis of the extracts. Thus, dichloromethane was selected, as the extraction solvent, over water or alcohols. Hence, the observed promising activities of the extract in reducing the colitis severity can be attributed to the non-polar constituents. These fat-soluble constituents are likely to be the sulfur-containing compounds which are found excessively in the essential oil of onion.

5- Herbs extracts contain various active ingredients like flavonoids and etc., the authors should be specified that which compound is responsible for the observed modulatory results in their investigation.

Response: It is true that various active ingredients do present in herb extracts, however, most of the documented anti-inflammatory activities shown by onion are attributed to the known flavonoid quercetin, where polar solvents, particularly water and/or alcohols, were used to prepare the tested extracts. However, to fulfill this study aim; to investigate the possibility of having other ingredients with promising activities, particularly in reducing the colitis severity, the applied extraction method (using the nonpolar solvent, dichloromethane) in this study was selected to provide extracts with, mainly, non-polar constituents, which, in a way, resembles the onion essential oils, where the highly polar bioactive flavonoids, including quercetin, are not among them. Hence, the observed promising activities of the extract in reducing the colitis severity can be attributed to the non-polar constituents. These fat-soluble constituents are likely to be the sulfur-containing compounds which are found excessively in the essential oil of onion. Our ultimate goal is to unambiguously identify the active ingredient(s) in this extract, however, this is a whole different project and is beyond the scope of our current study.

The above paragraph was added to the discussion section to justify the use of dichloromethane over water and/or alcohols.

6- The results of OBE 200 mg/kg impact on pro-inflammatory molecules are not shown in the Figures 7 & 8 for preventative approach.

Response: we now included IF analysis for the various tested pro-inflammatory molecules using the 200 mg/kg dose for the preventative approach in the revised version of the manuscript (NEW Figs 7 and 8). We also included western blotting analysis of the tested molecules using 100 and 200 mg/kg doses of OBE for the preventative and treatment approaches in the revised version of the manuscript (NEW Fig 9 A). 

7- Sample size estimation is not clarified. The number of mice are varied between examined groups and may impact on the overall conclusion. Why we had more animal subjects (almost double) in 100 OBE mg/kg group than the 200 OBE mg/kg group?

Response: we included large number of mice in all groups and we were including the 100 mg/kg dose as (positive control) in each experiment (based on our previously published report using the prophylactic approach, Ref 25). In general, there is minute variability between individual mice in each group, which is confirmed by tight range of SEM in the figures. 

8- For a better conclusion, Comparative analysis between two different OBE doses in all of the examinations is needed.

Response: we now included IF analysis for the various tested pro-inflammatory molecules using the 200 mg/kg dose for the preventative approach in the revised version of the manuscript (NEW Figs 7 and 8). We also included western blotting analysis of the tested molecules using 100 and 200 mg/kg doses of OBE for the preventative and treatment approaches in the revised version of the manuscript (NEW Fig 9 A). 

A comparative analysis is now included for these two doses of OBE using IF and western blotting analysis. There is no further enhancement in the anti-inflammatory effects of OBE using the 200 dose when compared to the 100 mg/kg dose. 

Reviewer #2: 

The manuscript by Dr. Maitham Khajah et al., suggests that Onion bulb extract (OBE) can reverse and prevent colitis in mice via inhibition of pro-inflammatory signaling molecules and neutrophil activity. The authors demonstrated that OBE treatment significantly reduced colitis severity, the colonic expression/activity profile of pro-inflammatory molecules, inhibited neutrophils-induced superoxide release, and increased spontaneous apoptosis of neutrophils in vitro. The data demonstrated the anti-inflammatory properties of onion bulb extract (OBE) in reducing colitis severity in mice. The study is well designed and logical but partially supported by the data. The experiments are well developed, need some in-vivo effector site protection of colon as compared to bone marrow-derived induced neutrophils function. The specific comments are:

1. The figure shows the effects of OBE on DSS induced colitis. The changes in colon infiltrating cells is crucial for any effector function is missing. Further, it is not clear that what is the maximum effective dose as 100 or 200 mg/kg body weight.

Response: we are now providing western blotting analysis of the tested pro-inflammatory molecules and immune cell markers (CD3; T-cells, F4/80; macrophages, Ly6G; neutrophils, and CD19; B-cells) for 100 and 200 mg/kg doses of OBE in the revised version of the manuscript (new Figs 7, 8, and 9 A). We showed that the expression level of neutrophils (but not macrophage) marker was significantly increased in DSS-treated mice and was reduced by OBE treatment using the preventative and treatment approaches. Both T- and B-cell markers were not detected by western blotting analysis even when very high concentration of primary or secondary antibodies were used (data not shown). 

In regards to the macroscopic assessment, 200 mg/kg was more potent in reducing colitis severity parameters when compared to the 100 mg/kg using both approaches (Tables 1 and 2). In regards to the microscopic assessment, there is slight improvement with the 200 mg/kg (only with the preventative but not the treatment approach) when compared with the 100 mg/kg. 

In general, 100 mg/kg dose is the maximum effective dose of OBE since no significant additional improvement was observed by the 200 mg/kg dose

2. The percentage of changes in body weight data is very nice. It is not clear as to which immune cells recruited to the colon and OBE treatment reduce/enhances those immune populations.

Response: we are now providing western blotting analysis of the immune cell markers for 100 and 200 mg/kg doses of OBE in the revised version of the manuscript (NEW Fig 9 A). We showed that the expression level of neutrophils (but not macrophage) marker was significantly increased in DSS-treated mice and was reduced by OBE treatment using the preventative and treatment approaches. Both T- and B-cell markers were not detected by western blotting analysis even when very high concentration of primary or secondary antibodies were used (data not shown). 

3. Need a strong rationale for using bone marrow-derived neutrophils in place of neutrophils present in the colon during DSS and OBE treatment.

Response: this is a very valid comment, and we choose to use naïve bone-marrow-derived neutrophils rather than neutrophils present at the site of inflammation (colon) for the following reasons: a) to perform the in vitro assays we need very large number of cells which can be obtained easily from the bone marrow but not from peripheral organs (e.g. colon) or the blood. b) most immune cells present at inflammatory site are already primed/desensitized due to the presence of large amounts of pro-inflammatory molecules which might affect their function/behavior ex-vivo. c) we wanted to test these cells specifically in response to stimulation with a bacterially-derived stimulus WKYMVm (fMLP-peptide) since bacteria/bacterial products are considered one of the main driving force for colitis in humans and various animal models. d) we already used bone-marrow derived neutrophils previously for various in vitro assays linked to colitis, please see the below published manuscripts from our group: 

PMID: 28487234, PMID: 23355730, PMID: 21393420, PMID: 20872834, PMID: 27997590 

4. Histological data at both doses (100 or 200 mg/kb) shows similar results. I suggest PI to just follow one dose for clarity in this manuscript.

Response: we agree with the reviewer comment regarding the similarity in the response obtained between these two doses at the histological level, but we thought to keep both doses since there is significant difference at the macroscopic level (Tables 1 and 2). 

5. The observation of changes in COX-2, P-AKT, P ERK1/2 through immunofluorescence analysis is not clear and needs a better technique like western for a prudent conclusion.

Response: we are now providing western blotting analysis of the tested molecules for 100 and 200 mg/kg doses of OBE in the revised version of the manuscript (NEW Fig 9 A). 

6. Fig.7 is a very poor quality presentation. It is not clear as cells are making these or a just antibodies background. Need some better techniques like immunocytochemistry (works better in paraffin section).

Response: we are now providing western blotting analysis of the tested molecules which gives better quantification analysis of these proteins (NEW Fig 9 A).

---

## [Decision Letter · Decision Letter 1]

13 Oct 2020

Onion bulb extract can both reverse and prevent colitis in mice via inhibition of pro-inflammatory signaling molecules and neutrophil activity

PONE-D-20-14395R1

Dear Dr. Khajah,

We’re pleased to inform you that your manuscript has been judged scientifically suitable for publication and will be formally accepted for publication once it meets all outstanding technical requirements.

Kind regards,

Mathilde Body-Malapel

Academic Editor

PLOS ONE

Reviewers' comments:

Reviewer's Responses to Questions

**Comments to the Author**

1. If the authors have adequately addressed your comments raised in a previous round of review and you feel that this manuscript is now acceptable for publication, you may indicate that here to bypass the “Comments to the Author” section, enter your conflict of interest statement in the “Confidential to Editor” section, and submit your "Accept" recommendation.

Reviewer #1: All comments have been addressed

2. Is the manuscript technically sound, and do the data support the conclusions?

Reviewer #1: Yes

3. Has the statistical analysis been performed appropriately and rigorously? 

Reviewer #1: Yes

4. Have the authors made all data underlying the findings in their manuscript fully available?

Reviewer #1: Yes

5. Is the manuscript presented in an intelligible fashion and written in standard English?

Reviewer #1: Yes

6. Review Comments to the Author

Reviewer #1: (No Response)

7. PLOS authors have the option to publish the peer review history of their article (what does this mean?). If published, this will include your full peer review and any attached files.

Reviewer #1: **Yes: **Ehsan Gharib

---

## [Editor Report · Acceptance letter]

15 Oct 2020

PONE-D-20-14395R1 

Onion bulb extract can both reverse and prevent colitis in mice via inhibition of pro-inflammatory signaling molecules and neutrophil activity 

Dear Dr. Khajah:

I'm pleased to inform you that your manuscript has been deemed suitable for publication in PLOS ONE. Congratulations! Your manuscript is now with our production department. 

Kind regards, 

on behalf of

Dr. Mathilde Body-Malapel 

Academic Editor

PLOS ONE